# Biodiversity can benefit from climate stabilization despite adverse side effects of land-based mitigation

Haruka Ohashi [1*], Tomoko Hasegawa [2,3], Akiko Hirata [1,4], Shinichiro Fujimori [3,5,6], Kiyoshi Takahashi [3], Ikutaro Tsuyama[7], Katsuhiro Nakao[8], Yuji Kominami[9], Nobuyuki Tanaka[10], Yasuaki Hijioka[4] & Tetsuya Matsui [1]

Limiting the magnitude of climate change via stringent greenhouse gas (GHG) mitigation is necessary to prevent further biodiversity loss. However, some strategies to mitigate GHG emission involve greater land-based mitigation efforts, which may cause biodiversity loss from land-use changes. Here we estimate how climate and land-based mitigation efforts interact with global biodiversity by using an integrated assessment model framework to project potential habitat for five major taxonomic groups. We find that stringent GHG mitigation can generally bring a net benefit to global biodiversity even if land-based mitigation is adopted. This trend is strengthened in the latter half of this century. In contrast, some regions projected to experience much growth in land-based mitigation efforts (i.e., Europe and Oceania) are expected to suffer biodiversity loss. Our results support the enactment of stringent GHG mitigation policies in terms of biodiversity. To conserve local biodiversity, however, these policies must be carefully designed in conjunction with land-use regulations and societal transformation in order to minimize the conversion of natural habitats.

[1] Center for International Partnerships and Research on Climate Change, Forestry and Forest Products Research Institute, Forest Research and Management Organization, Matsunosato 1, Tsukuba, Ibaraki 305-8687, Japan. [2] Department of Civil and Environmental Engineering, College of Science and Engineering, Ritsumeikan University, 1-1-1 Nojihigashi, Kusatsu, Shiga 525-8577, Japan. [3] Center for Social and Environmental Systems Research, National Institute for Environmental Studies, Japan, 16-2 Onogawa, Tsukuba, Ibaraki 305-8506, Japan. [4] Center for Climate Change Adaptation, National Institute for Environmental Studies, 16-2 Onogawa, Tsukuba, Ibaraki 305-8506, Japan. [5] Department of Environmental Engineering, Graduate School of Engineering, Kyoto University, 361, C1-3, Nishikyo, Kyoto, Kyoto 615-8540, Japan. [6] Energy Program, International Institute for Applied System Analysis, Schlossplatz 1, A-2361 Laxenburg, Austria. [7] Hokkaido Research Center, Forestry and Forest Products Research Institute, Forest Research and Management Organization, Hitsujigaoka 7, Toyohira, Sapporo, Hokkaido 062-8516, Japan. [8] Kansai Research Center, Forestry and Forest Products Research Institute, Forest Research and Management Organization, Nagai-kyutaro 68, Momoyama, Fushimi, Kyoto, Kyoto 612-0855, Japan. [9] Department of Disaster Prevention, Meteorology and Hydrology, Forestry and Forest Products Research Institute, Forest Research and Management Organization, Matsunosato 1, Tsukuba, Ibaraki 305-8687, Japan. [10] Faculty of International Agriculture and Food Studies, Tokyo University of Agriculture, Sakuragaoka 1-1-1, Setagaya, Tokyo 156-8502, Japan. *email: oharu0429@gmail.com

Over the last century, anthropogenic interventions in natural ecosystems have caused an exceptionally rapid loss of biodiversity. According to the IUCN Red List, 198 vertebrates have been verified as 'extinct' since 1900, and the average rate of vertebrate species loss over the last century was 22 times higher than in the pre-anthropogenic era[1,2]. Land-use change has been the largest driver of this biodiversity loss[1,3]. In particular, the expansion of agricultural area to support an increasing global population has caused major ecosystem changes over millenia[1]. While the global population increased from 1650 million in 1900[4] to 6145 million in 2000[5], agricultural area increased from 21 million km² in 1900 to 50 million km² in 2000, covering about 35% of the Earth's land surface[6]. Climate change is also becoming a major threat to biodiversity[3,7,8]. Since 1880, there has been an average warming of 0.85 °C globally[9], and many organisms are likely changing their distributions as a means of adapting to climate change[10].

With an increasing recognition of the importance of biodiversity for human society[11], preventing further biodiversity loss is now a target of global sustainability policy, such as the Aichi Targets of the Convention on Biological Diversity and Sustainable Development Goals of the United Nations Development Programme. Although these political frameworks are focused on the near future, it is also important to take a longer perspective along with these policy frameworks in order to project long-term global sustainable development. As we move towards a global population of 9–10 billion people by 2050, striking a balance between competing demands for land to provide goods and services (e.g., food, water, timber, energy, settlements and recreation) and safeguarding Earth's life-support system will become more critical issues as we aim to achieve sustainable development[12]. Increasing demand for agricultural goods[13] may put pressure on future biodiversity. For example, in newly developed socioeconomic scenarios, so-called shared socioeconomic pathways (SSPs), global cropland is projected to expand by 25% and pasture by 6% by 2100 (relative to 2010) in the middle-of-the-road scenario (SSP2), while aggregated area of forest and other natural land is projected to decrease by 6.1 million km² by 2100 compared to 2010[14].

Furthermore, as the global temperature rises due to climate change, biodiversity loss is expected to become more severe[7,8]. Previous studies have agreed that reducing the degree of climate change by stringent greenhouse gas (GHG) mitigation activities can prevent a substantial loss of biodiversity[7,8]. However, most of these studies only consider the effects of change in climatic condition[15]. Recently, integrated assessment models revealed that the most stringent GHG mitigation scenarios require substantial land-based mitigation options such as large-scale bioenergy crop production, afforestation, and avoiding deforestation[16,17], which would lead to a wide range of possible changes in land use[18,19]. One study's findings suggested synergy between land-use change for GHG mitigation and biodiversity conservation[20], whereas others showed that potential land-use changes for GHG mitigation may cause further biodiversity loss[21,22]. In reality, land-use change and climate change occur simultaneously, and losses or gains of suitable habitats occur not only due to land-use change or climate change, but also due to their combined effects[23,24]. Therefore, integrated assessment of climate change and land-use change is urgently needed to clarify whether climate change mitigation measures truly contribute to biodiversity conservation.

Here, we describe the combined effects of climate change and land-use change on biodiversity under ambitious climate change mitigation efforts at a global scale. We separately account for the beneficial effect of limiting the magnitude of climate change and the effect of additional land-use change, which can be either destructive or preventative. We use a one-way economy-land-biodiversity modelling framework. First, we project regional aggregated land use using the Asia-Pacific Integrated Model/Computable General Equilibrium Model (AIM/CGE)[25], which was downscaled to high spatial resolution with the AIM/PLUM model[26]. The AIM/CGE implements climate change mitigation in the form of a global uniform carbon tax on GHG emissions from the agriculture, land-use, and energy sectors. Then, we estimate the losses and gains in suitable habitat for 8428 species in five taxonomic groups (vascular plants, amphibians, reptiles, birds, and mammals) by using a species distribution model based on occurrence–environment correlations. Although projecting future suitable habitats is relatively simple, whether species can occupy a space or not relies on their dispersal abilities[27]. In this study, we incorporate variation of dispersal ability at the species level into the model by using a simple approach based on life-history traits, although more information is needed to represent all of the complex mechanisms related to dispersal[27] (see Methods).

We consider two future scenarios: baseline versus mitigation scenarios. The baseline scenario assumes no GHG emission reductions, and the global mean surface temperature continues to rise. In contrast, the mitigation scenario assumes the ambitious climate change mitigation efforts of the 2 °C scenario, where GHG emission levels are cut to RCP2.6 emission levels with a residual climate change impact. We consider five alternative scenarios for land-use change that reflect different future socioeconomic development trends, which correspond to five SSPs[28,29]. Consequently, the total areas of forest (including afforestation) and bioenergy crops are larger in the mitigation scenario than in the baseline scenario, whereas the total areas of cropland (excluding bioenergy crops), pasture, and other natural land are smaller[26]. For climate information, we use variables based on five General Circulation Models (GCMs) of two representative concentration pathways: RCP2.6, which roughly corresponds to a global mean temperature rise from preindustrial times of <2 °C by 2100[30], for the mitigation scenario, and RCP8.5, which has a 2.6–4.8 °C rise[31], for the baseline scenario.

Moreover, to decompose the individual effects of land-use change and climate change, we run three hypothetical scenarios that were produced by combining land-use change and climate change scenarios (Supplementary Table 1). Finally, we analyse the individual effects of land-use change and climate change on loss and gain of species' suitable habitat and compare these areas between the mitigation and baseline scenarios (see Methods).

We conclude that climate stabilization by stringent GHG mitigation, in general, can provide a net benefit to global biodiversity despite some regions suffering a loss of biodiversity due to land-based GHG mitigation efforts. These beneficial effects become more relevant in the latter half of this century, rather than in the near future. To enhance the synergy between GHG mitigation and the prevention of biodiversity loss, strong land-use regulations and societal transformations that foster sustainability will be necessary.

## Results

**Factors driving losses and gains of suitable habitats.** Our results indicate that the total global area of suitable habitats cannot avoid some degree of loss regardless of whether climate change mitigation is implemented or not (Fig. 1). However, less suitable habitat was lost in the mitigation scenario compared to the baseline scenario for all taxonomic groups. This trend can be seen in projections for both the 2050s and 2070s. In the 2050s, losses of suitable habitat were 12.6–23.3% of current suitable habitat area in the mitigation scenario and 14.9–26.0% in the baseline scenario. In the 2070s, losses of suitable habitat were 16.8–27.8% of current suitable habitat area in the mitigation scenario and 21.6–35.9% in the baseline scenario. The second important

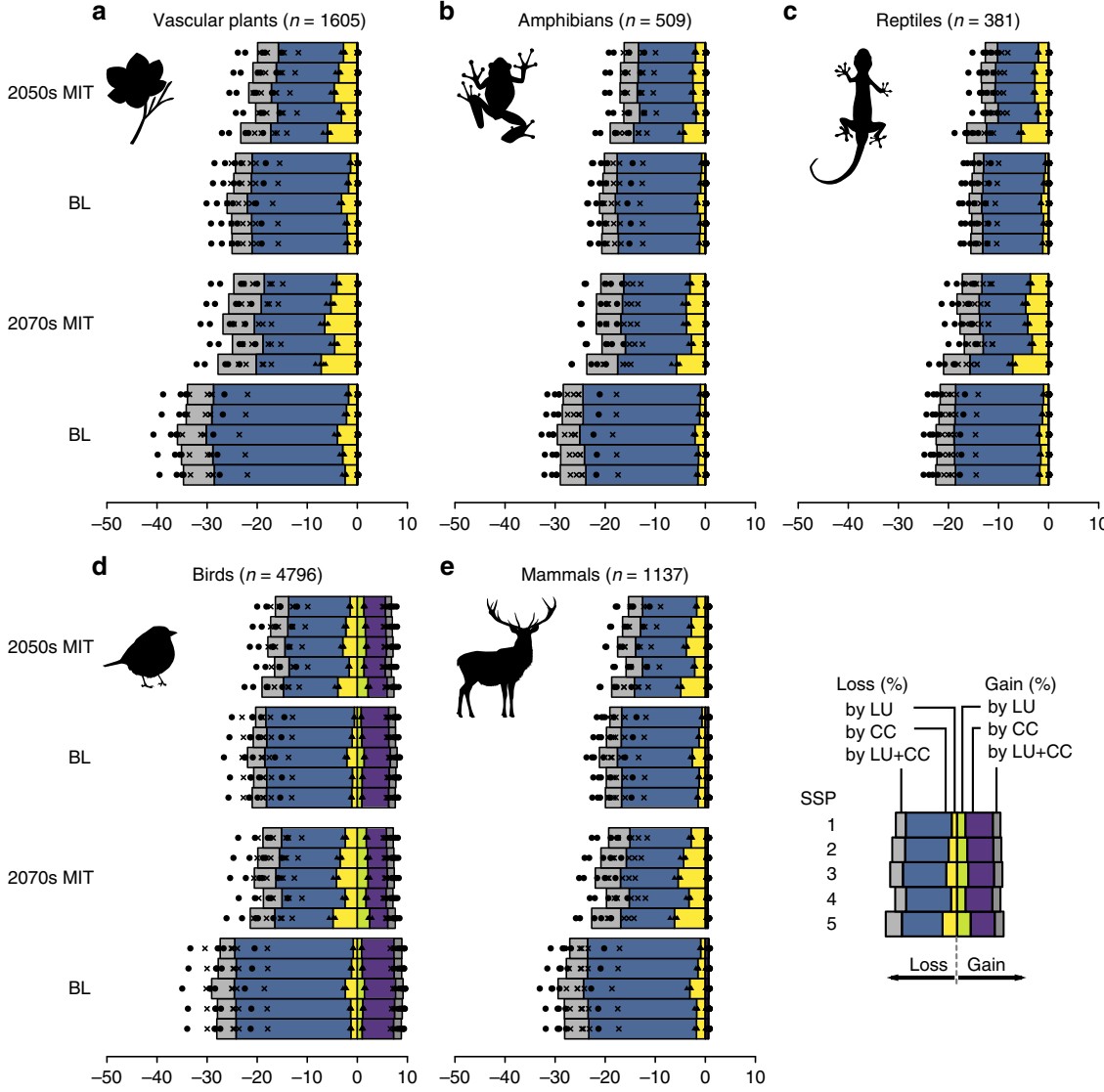

**Fig. 1** Proportion of losses and gains in suitable habitat from the present to the 2050s and 2070s in mitigation (MIT) and baseline (BL) scenarios. Individual contributions of land-use change, climate change, and combined effect to losses and gains in suitable habitat were identified. Figure shows average proportion of five GCMs in each SSP, for each taxonomic group: **a** vascular plants, **b** amphibians, **c** reptiles, **d** birds, and **e** mammals. Corresponding data points for the contribution of land-use change are represented as triangles, the cumulative contributions of land-use change and climate change are represented as crosses, and the cumulative contributions of all drivers are represented as circles. Source data are provided as a Source Data file

finding is that the differences between mitigation and baseline scenarios became larger over time: every taxonomic group showed larger habitat losses in the 2070s than in the 2050s.

Climate change was always the dominant factor compared to land-use change, although the magnitude of these factors differed across scenarios. Not surprisingly, the loss of suitable habitat due to climate change was smaller in the mitigation scenario than in the baseline scenario (Fig. 1). On the other hand, loss of suitable habitat due to land-use change was larger in the mitigation scenario than in the baseline scenario. A larger proportion of land was converted in the mitigation scenario, with increased areas for afforestation and bioenergy crop production (Supplementary Fig. 2). Because the effect of climate change was larger than that of land-use change in all taxonomic groups, the baseline scenario is consequently projected to have a more negative impact in total.

Additionally, we found a significant difference in loss and gain of suitable habitat among SSPs (Tables 1 and 2). SSP1 showed the smallest loss and relatively large gain in suitable habitat.

SSP4 showed a relatively small loss and small gain of suitable habitat, whereas SSP3 and SSP5 showed a relatively large loss and large gain of suitable habitat.

The difference in losses of suitable habitat among taxonomic groups was also significant: vascular plants showed the largest loss, followed by birds, mammals, amphibians, and reptiles (Tables 1 and 2). Furthermore, gains of suitable habitat were dependent on the taxonomic group. Those containing species with a strong dispersal ability had modest gains in suitable habitat (mammals: 0.5–0.7%, birds: 6.8–9.1%), whereas the other three groups hardly gained new suitable habitat (Fig. 1, Table 2). Moreover, for mammals and birds, the gains were much lower than losses in all years and scenarios (Fig. 1).

**Regional variation in the net benefit of the mitigation.** Losses of suitable habitat also varied across the native regions of taxa. Total losses in the baseline scenario were relatively large for species in Europe, Africa, and South America (Fig. 2). For most regions,

**Table 1 Result of the GLMM for area of lost suitable habitat**

| Parameter | Estimate | 95% CI | z | p |
|---|---|---|---|---|
| (Intercept) | −1.757 | −1.816 to −1.699 | −58.977 | <0.001 |
| Scenario (MIT)[a] | −0.103 | −0.107 to -0.098 | −43.798 | <0.001 |
| Year (2070s)[b] | 0.366 | 0.362 to 0.371 | 157.261 | <0.001 |
| SSP (SSP1)[c] | −0.046 | −0.051 to −0.040 | −17.493 | <0.001 |
| SSP (SSP3)[c] | 0.043 | 0.038 to 0.048 | 16.508 | <0.001 |
| SSP (SSP4)[c] | −0.037 | −0.042 to −0.032 | −14.364 | <0.001 |
| SSP (SSP5)[c] | 0.107 | 0.102 to 0.112 | 40.967 | <0.001 |
| Taxonomic group (Amphibians)[d] | −0.331 | −0.450 to −0.213 | −5.478 | <0.001 |
| Taxonomic group (Reptiles)[d] | −0.636 | −0.769 to −0.503 | −9.393 | <0.001 |
| Taxonomic group (Birds)[d] | −0.320 | −0.387 to −0.253 | −9.344 | <0.001 |
| Taxonomic group (Mammals)[d] | −0.332 | −0.422 to −0.241 | −7.199 | <0.001 |
| GCM (HadGEM2-ES)[e] | −0.060 | −0.065 to −0.054 | −22.675 | <0.001 |
| GCM (GFDL-CM3)[e] | 0.160 | 0.155 to 0.166 | 60.622 | <0.001 |
| GCM (MIROC-ESM-CHEM)[e] | 0.138 | 0.133 to 0.144 | 52.304 | <0.001 |
| GCM (NorESM1-M)[e] | −0.242 | −0.248 to −0.237 | −92.188 | <0.001 |
| Scenario (MIT)[a] × Year (2070s)[b] | −0.128 | −0.134 to −0.121 | −38.760 | <0.001 |

Parameter estimates with their associated 95% confidence interval and test statistics (Wald's z-score and p values for Wald test) of the GLMM for area of lost suitable habitat. The shape parameter of the Gamma distribution was estimated as 1.756. Standard deviation of random effects was estimated as 1.186. [a]Baseline (BL) scenario was set as the reference. [b]The years of the 2050s were set as the reference. [c]SSP2 was set as the reference. [d]Vascular plants was set as the reference. [e]IPSL-CM5A-LR was set as the reference

**Table 2 Result of the GLMM for area of gained suitable habitat**

| Parameter | Estimate | 95% CI | z | p |
|---|---|---|---|---|
| (Intercept) | −26.639 | −26.815 to −26.463 | −296.805 | <0.001 |
| Scenario (MIT)[a] | 0.024 | 0.018 to 0.029 | 9.012 | <0.001 |
| Year (2070s)[b] | 0.027 | 0.022 to 0.032 | 10.234 | <0.001 |
| SSP (SSP1)[c] | 0.007 | 0.002 to 0.013 | 2.570 | 0.010 |
| SSP (SSP3)[c] | 0.005 | −0.000 to 0.011 | 1.825 | 0.068 |
| SSP (SSP4)[c] | −0.014 | −0.019 to −0.008 | −4.661 | <0.001 |
| SSP (SSP5)[c] | 0.038 | 0.032 to 0.044 | 13.015 | <0.001 |
| Taxonomic group (Amphibians)[d] | 0.676 | 0.317 to 1.034 | 3.696 | <0.001 |
| Taxonomic group (Reptiles)[d] | 0.568 | 0.166 to 0.969 | 2.771 | 0.006 |
| Taxonomic group (Birds)[d] | 23.218 | 23.015 to 23.421 | 224.049 | <0.001 |
| Taxonomic group (Mammals)[d] | 2.855 | 2.582 to 3.128 | 20.494 | <0.001 |
| GCM (HadGEM2-ES)[e] | 0.000 | −0.006 to 0.006 | −0.077 | 0.939 |
| GCM (GFDL-CM3)[e] | −0.011 | −0.017 to −0.006 | −3.906 | <0.001 |
| GCM (MIROC-ESM-CHEM)[e] | 0.102 | 0.096 to 0.108 | 34.644 | <0.001 |
| GCM (NorESM1-M)[e] | 0.028 | 0.022 to 0.033 | 9.420 | <0.001 |
| RCP (MIT)[a] × Year (2070s)[b] | −0.027 | −0.034 to −0.019 | −7.229 | <0.001 |

Parameter estimates with their associated 95% confidence interval and test statistics (Wald's z-score and p values for Wald test) of the GLMM for area of gained suitable habitat. The shape parameter of the Gamma distribution was estimated as 1.405. Standard deviation of random effects was estimated as 3.593. [a]Baseline (BL) scenario was set as the reference. [b]The years of the 2050s were set as the reference. [c]SSP2 was set as the reference. [d]Vascular plants was set as the reference. [e]IPSL-CM5A-LR was set as the reference

total losses of suitable habitat drastically decreased in the mitigation scenario (Figs 2 and 3). However, in the 2050s, some scenario–taxon combinations in Europe and Oceania were predicted to experience further loss of suitable habitat in the mitigation scenario. These regional variations in habitat losses can be explained by the magnitude of land-use and climate changes for each region. When looking at the individual effect of land-use change on species' suitable habitat loss, the magnitude of loss was correlated with the proportion of land that has changed from the current to another land use (Fig. 4a). Similarly, the proportion of habitat loss due to climate change was correlated with the degree of change in the maximum temperature of the warmest month, one of the 19 bioclimatic variables used to construct models (Fig. 4b). In the regions predicted to have large land-use changes in the mitigation scenario (e.g., Europe), only a small reduction in the total loss of suitable habitat was achieved by mitigation due to the increased loss of suitable habitat by land-use change (Figs 3 and 4).

## Discussion

Our results highlight the fact that climate change mitigation can reduce the risk of species loss at a global scale, but we must consider the effects of land-use change associated with GHG mitigation efforts. Certain losses of suitable habitat were inevitable under both mitigation and baseline scenarios. This is partly due to the nature of the climate characteristics, because even if GHG emissions were immediately halted, temperature rise could continue until the middle of the century. When looking at the individual effect of the two drivers, climate change had marked effects on loss/gain of suitable habitat regardless of differences in future socioeconomic development trends (Fig. 1), which was consistent with previous studies at a larger spatial scale[32]. The spatial extent of each driver differs: climate change affects all of the land surfaces, whereas land-use change occurs locally. Even in the 2070s, >70% of the terrestrial land surface was unchanged in both scenarios (Supplementary Fig. 2). Therefore, although stringent mitigation measures have been implemented,

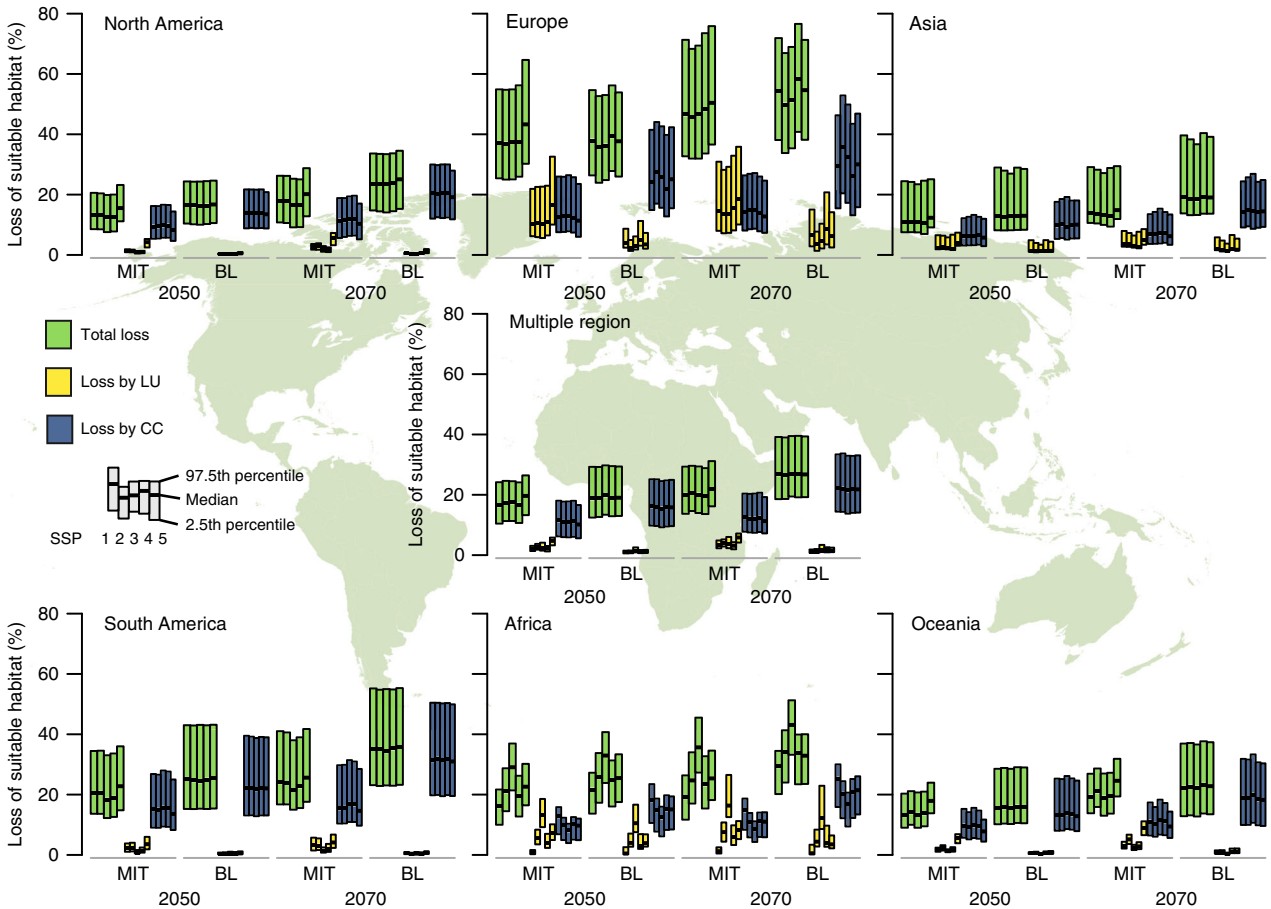

**Fig. 2** Proportion of total loss, loss due to land-use change, and loss due to climate change in suitable habitat from the present to the 2050s/2070s in mitigation (MIT) and baseline (BL) scenarios, aggregated by species' native region. Each box represents 2.5 and 97.5 percentiles of the mean proportion of each combination of taxonomic group and GCM within each SSP. The bold line in each box shows the median value. Source data are provided as a Source Data file. World map was generated by using software QGIS ver 2.18.27 and polygon data obtained from http://www.iucnredlist.org/

substantially more area can be affected by the residual impacts of climate change relative to that of land-use change.

Note, however, that our results may underestimate the impact of land-use change for two reasons. First, our models focus on land-cover change and are based on the assumption that the effect of land-use intensity within each land-use type does not affect species' habitat suitability. For example, all of the five SSP scenarios more or less estimate an increase in crop yield in the future resulting from technological development and irrigation expansion[33], as reflected in differences in the expansion rates of agricultural land use in this study (i.e., if higher yield is expected, then extension of agricultural land use could be limited). However, increasing crop yield through agricultural practices (e.g., significant input of inorganic fertilizer or pesticide) may add further risks for biodiversity loss[34] and require additional efforts for conservation[35]. Although not all technologies that aim to achieve crop yield increases cause negative impacts on biodiversity[36,37], the lack of this factor in our species distribution model may have led to underestimated biodiversity losses from land-use change. As another example, the negative impact of land-use change associated with GHG emission reduction on biodiversity within the forest may become large if the afforestation is conducted with non-native species and without efforts to restore the environment of the original natural forest[21,22]. These changes in habitat suitability arising from differences in environmental quality may cause systematic bias in our analysis. Second, our analyses only evaluated those species with enough

data to construct a species distribution model, meaning that species adapted to a long history of human land use were likely selected. As a result, the effect of land-use change on the species vulnerable to human disturbance might be underestimated in our study.

The beneficial effects of GHG mitigation become more relevant in the long term, rather than in the near future (Figs 1–3), which is due to the delayed climate response to a reduction in GHG emissions. These results have significant implications for policy-making in the sense that focusing only on the near term provides limited information to appropriately assess future comprehensive biodiversity situations. Despite the limited information for a much longer time scale (e.g., more than a century), the future climate effects can be much stronger drivers of biodiversity than land-use change, which highlights the need for stringent GHG mitigation actions.

Note that our results are highly dependent on assumptions regarding socio-economic conditions such as lifestyle[38] and technological development related to energy efficiency[39] and biomass utility[40], which have the potential to affect future land-use changes. These aspects are partially expressed as a difference among SSP scenarios. Because SSPs involve changing several factors simultaneously and it is impossible to identify the factor that determines land-use change, several aspects of the land-use storyline in SSP1 (e.g., strong land-use change regulation, increasing crop yields, low animal-calorie shares, and low waste)[18] may be the candidate feature that contributed to minimize

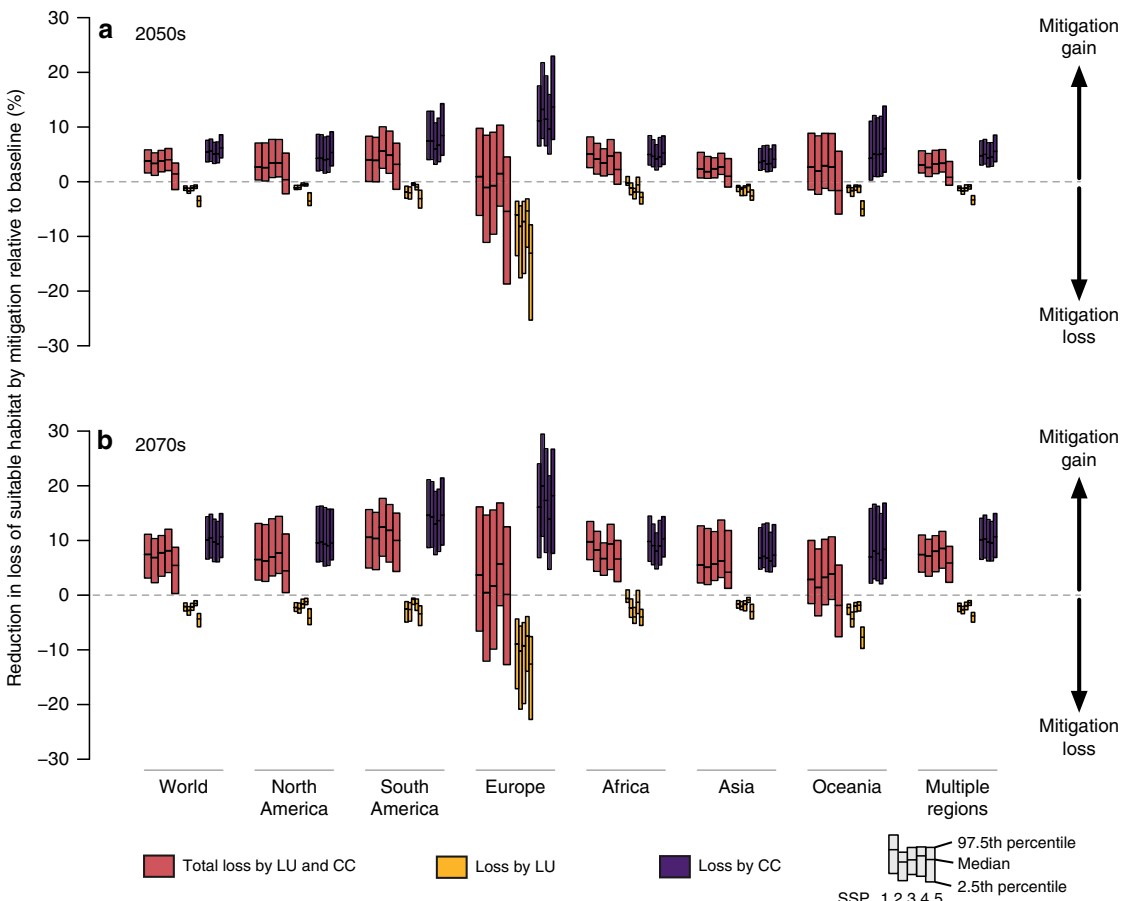

**Fig. 3** Net benefit of mitigation policy: reduction of proportion of loss of suitable habitat from the present to the 2050s(upper)/2070s(lower) by mitigation relative to baseline, aggregated by species' native region. Each box represents 2.5 and 97.5 percentiles of the mean proportion of each combination of taxonomic group and GCM within each SSP. The bold line in each box shows the median value. Source data are provided as a Source Data file

conversion of forest and other natural land and prevent the loss of suitable habitats. For example, the beneficial effects of GHG mitigation on biodiversity become small in the scenario with higher agricultural demand induced by bioenergy consumption (Figs 1–3, Supplementary Figs 7 and 8). This result indicates that reducing society's total energy demand may enhance synergies between GHG mitigation and the prevention of biodiversity loss. Additionally, improving yields on existing croplands by environmentally sustainable intensification techniques[36,37] and changing our diet[41] might be another option to reduce the additional pressure on biodiversity resulting from GHG mitigation activity. Currently, widely used land-use predictions are generated for single RCP-SSP combinations, which does not allow isolation of pure differences among socio-economic conditions[23]. Further work that explicitly considers various socio-economic conditions and GHG mitigation options will need to specify the critical socio-economic factor necessary to achieve both GHG mitigation and biodiversity targets.

In the regions that experienced large land-use changes, some species may lose suitable habitat due to changes in land use for GHG mitigation (Figs 3 and 4). Even if suitable habitat was artificially created by land-based mitigation, many organisms with a poor dispersal ability would not immediately disperse into the new habitat (vascular plants, amphibians, and reptiles in this study; Fig. 1). We note that our results only shed light on suitable habitat area, which is only one target of GHG mitigation. Future research should include various sectors, such as human health and food security. However, our findings indicate that giving

additional conservation effort to species vulnerable to land-use change may be necessary to balance GHG mitigation and prevent further loss of biodiversity.

Our research highlights the importance of considering the impact of land-use change caused by GHG mitigation activities as well as climate change. More importantly, there needs to be careful planning to achieve synergies between GHG mitigation and the prevention of biodiversity loss. In the political arena, climate and biodiversity discussions are currently divided; these two policy frameworks should be integrated or at least communicated properly. Furthermore, integrated modelling of the economy–land–biodiversity causal link will enable us to assess the direct and/or indirect effects of various socioeconomic conditions on biodiversity, which will encourage the exploration of pathways to achieve multiple sustainable development goals in the future.

## Methods

**Species distribution modelling by MaxEnt.** Species distribution models, which predict a species' probability of occurrence across a landscape, take a correlative modelling approach to species' occurrence–environment relationships. This approach has recently gained importance as a tool to assess the impact of environmental change on the distribution of organisms[42]. In this study, we employed MaxEnt v3.3[43], one of the most robust modelling approaches, especially for cases in which only presence data are available and absence data are difficult to collect[44].

In this study, we used species distribution models to project the current and future probability of occurrence of numerous species. This method consists of the following four phases (Supplementary Fig. 1): pre-processing of data, species level modelling of occurrence–environment relationships, estimating available habitat, and decomposing the driver of habitat loss/gain. We used two future scenarios (mitigation and baseline) to evaluate the integrated effects of climate change and

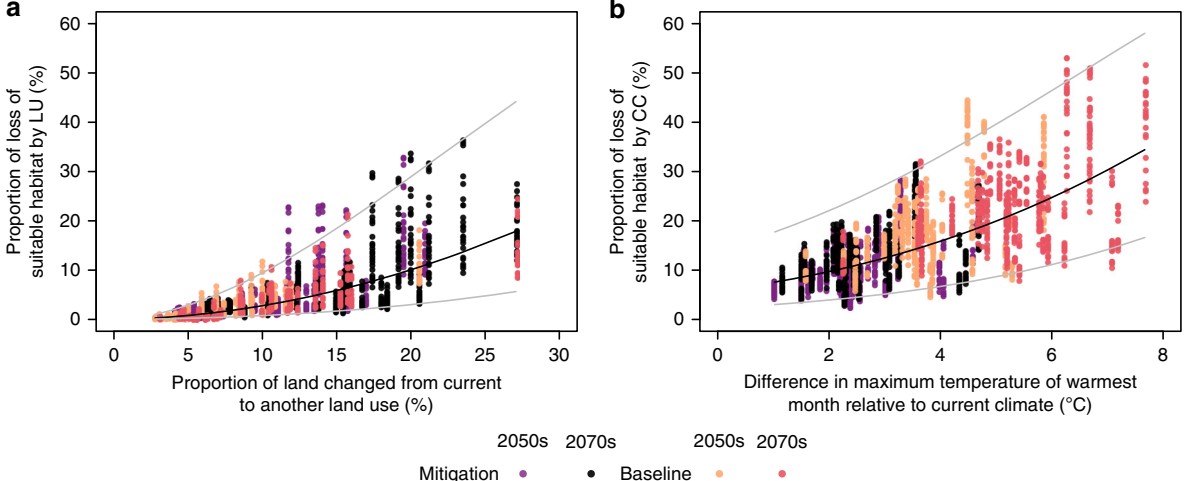

**Fig. 4** Effect of magnitude of land-use change and climate change on regional variation in loss of suitable habitat. **a** Effect of proportion of land that has changed from the current land use to another land use on proportion of loss in suitable habitat due to land-use change. Each point represents proportion of loss by land-use change, aggregated by species' native region and taxonomic groups for mitigation or baseline scenario in the 2050s and 2070s. Species in multiple regions were not used for the analysis. Regression lines (black) are based on the linear regression equation $Y = 0.14 + 1.68X$ (adjusted-$R^2$: 0.72); $Y$: logit transformed average proportion of loss of suitable habitat due to land-use change; $X$: logit transformed proportion of land that has changed from current to another land use. **b** Effect of difference in maximum temperature of warmest month (bio5) on proportion of loss in suitable habitat due to climate change. This variable was selected based on preliminary correlation analysis among 19 bioclimatic variables. Each point represents the average proportion of loss by climate change, aggregated by species' native region and taxonomic groups, for mitigation or baseline scenario in the 2050s and 2070s. Species in multiple regions were not used in the analysis. Regression lines are based on multiple linear regression of the equation $Y = -2.79 + 0.28X$ (adjusted-$R^2$: 0.41); $Y$: logit transformed average proportion of loss of suitable habitat due to climate change; $X$: difference in maximum temperature of warmest month (bio5). Grey lines indicate 95% prediction interval in each regression model. Source data are provided as a Source Data file

land-use change accompanied by stringent mitigation activity in the 2050 s and 2070 s. For each species, the areas of potential suitable habitat under the two future scenarios were calculated based on a species distribution model. To identify the drivers causing loss/gain of potential suitable habitat of each species under each scenario, we calculated the potential suitable habitats based on three hypothetical future scenarios (land-use change alone, climate change alone, and both). Finally, we compared the proportion of area loss/gain and the contribution of the two drivers (land-use change and climate change) between the mitigation and baseline scenarios. We used R ver. 3.3.2[45] and R packages rgdal 1.2–6[46], raster 2.5–8[47], rJava 0.9–8[48], dismo 1.1–4[49], and ENMeval 0.2.2[50] for the analyses.

**Filtering occurrence data from GBIF.** We used data from the Global Biodiversity Information Facility (GBIF)[51], which is the largest portal that collects distribution records of species based on digitized collection and survey data.

First, we obtained the entire GBIF dataset (as of July 2015) from the secretariat of GBIF. The data contained ~560 million records at this stage. From this, records of five major taxonomic groups (vascular plants, mammals, amphibians, reptiles, and birds) were extracted (first-order dataset). We then confirmed geographic location based on the latitude and longitude information; if the recorded country name and geographic location showed any discrepancy, those records were discarded. We also removed occurrence records with no location data and those that did not fall within land areas by referring to the GADM database of Global Administrative Areas[52] (second-order dataset), according to Warren et al.[53]. It is common that an occurrence–environmental relationship is not conserved outside of a species' native range[54]. Therefore, from the second-order dataset, we discarded records labelled as "fossil specimen," "unknown," or "living specimen" recorded outside of their native ranges (third-order dataset). To ascertain the species' native ranges in the third-order dataset, we referred to the IUCN Red List of Threatened Species[55] (http://www.iucnredlist.org/), which contains detailed information on the native ranges of 46,182 species (either continent or country level) for both non-threatened and threatened species. We matched the third-order dataset and the IUCN data and discarded those records with a discrepancy in location information between the two datasets. Then we labelled the native region as belonging to one of the following seven groups: North America, South America, Europe, Africa, Asia, Oceania, and multiple regions. Occurrence data in Antarctica were excluded from the analysis due to a lack of environmental data. As a result, our analysis is limited to the species already assessed by the IUCN Red List (fourth-order dataset). Finally, to reduce the effect of spatial clustering of records in the dataset[56,57], we confined the number of records per cell by subsampling the dataset. We sampled one occurrence record per 0.5 arc degrees grid (ca. 60 × 60 km at the equator) for all species (final dataset). Also, to avoid the effect of model inaccuracy from small sample size, we limited our analysis to species with ≥30 refined occurrence records[58].

Based on this data selection process, the total number of species in the final dataset was 8510. Species distribution models were constructed for these 8510 species. These presence data were used as the response variable for the construction of MaxEnt models for each species.

**Spatial bias of the database: generating background data.** A fundamental assumption of species distribution models is that the entire area of interest has been systematically or randomly sampled[59,60]. For example, by default, MaxEnt selects the background locations (sometimes called pseudo-absences) at the same probability across the target landscape and contrasts background against occurrence to estimate the relative probability of occurrence. However, in practice, occurrence records are likely to be spatially biased towards more easily accessed or better surveyed areas[59], which are affected by various social constraints[61,62]. When sampling is biased, the model cannot differentiate whether species are observed in particular environments because those locations are preferable or because they receive the largest search effort. This problem seriously affects the outcome for species distribution models derived from presence-only data and distorts our view of large-scale biodiversity patterns[56,59].

To overcome such problems and formulate models in geographic space, spatially-explicit information on sampling effort is required. However, as such information is often unavailable, methods to account for sampling bias are typically based on target group sampling[59]. Target group sampling uses the occurrence records of taxonomically related species observed by the same techniques as the focal species to estimate sampling. This method assumes that if the taxonomically related species have been observed in a survey, then the focal species also would have been observed there. To incorporate sampling bias into MaxEnt, we took the bias background approach, which uses the prior information on the spatial distribution of survey effort to preselect background locations before running MaxEnt. In this approach, the effect of sampling bias cancels out because it is common to both occurrence and background. To do this, we combined all occurrence records in the final dataset for each of the five taxonomic groups (including species with ≤30 records). We generated a set of background data for each of the five target groups, weighted by the sampling density of occurrence records[59]. In the model development procedure, the background data were extracted within the native range for each species.

**Explanatory variables used for modelling.** We used land-use variables and climatic variables as candidate explanatory variables. Land-use variables were obtained from the land-use allocation model of Hasegawa et al.[26]. For the analyses, the proportion of each of the five land-use types (cropland, pasture, forest, other natural land, and settled land) was stored in each grid cell at a spatial resolution of 0.5 arc degrees.

For the climatic variables, a dataset of monthly minimum temperature, maximum temperature, and precipitation was downloaded from WorldClim[63]. The dataset at a resolution of 30 arc seconds was averaged to values at a resolution of 0.5 arc degrees. Then, we calculated 19 bioclimatic variables based on the three climatic variables, and these bioclimatic variables were used for the model construction.

**Filtering candidate explanatory variables for modelling.** We tried to identify a species-specific set of explanatory variables for each of the 8510 species by using the following procedures. First, by using the five land-use and 19 bioclimatic variables, we generated all the possible combinations ($2^{24}$ = 16,777,216) of these variables. We then excluded explanatory variables showing collinearity[64]: (1) we excluded the set of explanatory variables that includes pairs of highly correlated variables (Pearson's product moment correlation coefficient ≥ |0.70|)[65,66], and (2) we excluded those explanatory variables for which the variance inflation factor (VIF) value, which indicates the degree of collinearity between two or more predictor variables[67], was ≥5. VIF values were calculated using "vif()" function of R package usdm 1.1–15[67].

In addition, to reduce computation time, we excluded the combination of nested subsets of other combinations, which is attainable from the subsequent regularization process in MaxEnt. After these procedures, there were 91 candidate combinations of parameters.

**Running MaxEnt.** MaxEnt is capable of building complex nonlinear functions of explanatory variables by combining simple mathematical transformations of explanatory variables, or so-called features[60]. In the present study, we used linear and quadratic features only, to avoid producing too complex a model, which could lead to extrapolation errors[44]. Within a given combination of explanatory variables, MaxEnt selects the features for each explanatory variable that contribute most to model fit using regularization[43,60]. Regularization is based on a combination of likelihood and a complexity penalty and reduces overfitting[60]. This process ensures that the model is not fit too precisely to the given dataset and removes unimportant features from the model. At the beginning, we set the regularization coefficient (i.e., β-multipliers) to 0, which indicates no regularization.

For each of the 8510 species, we developed a MaxEnt model for predicting the distribution probability by iteratively using the prepared 91 combinations of explanatory variables. To select the most parsimonious combination of explanatory variables, corrected Akaike information criterion (AICc)[68] values were compared among the 91 candidate models, and the model with the minimum value was selected. Next, we tested 31 regularization coefficients (from 0 to 15 at increments of 0.5) and choose the one for each species that maximizes model fit under the given combination of explanatory variables selected, based on AICc[69].

**Species selection by model performance evaluation.** Among the final models developed for the 8510 species, those with poor performance were discarded for subsequent analyses based on 10-fold cross validation. In this process, the occurrence and background datasets were divided randomly into 10 equal-sized groups, and models were built using $k - 1$ bins for calibration in each iteration (training set), with the left-out bin used for evaluation (test set). Model performance was assessed using the continuous Boyce index (CBI), which is used to evaluate model quality for predictions based on presence data only[70,71]. This index varies from −1 to 1; negative values indicate an incorrect model, values close to 0 mean a chance model, and positive values indicate a model whose predictions are consistent with the presence distribution in the test set[70]. In this study, models with CBI > 0 based on the 95% confidence interval were used for the subsequent analyses. After these procedures, final models for 8428 species were retained for subsequent analysis: 1605 vascular plants, 509 amphibians, 381 reptiles, 4796 birds, and 1137 mammals (Supplementary Data 1).

As a result of data requirements, our target species have a high proportion of common species with low extinction risk and a low proportion of endangered species (Supplementary Table 2). Additionally, areas with intensively surveyed regions (e.g., North America and Europe) had a high proportion of modelled species (Supplementary Table 2).

**Estimation of current suitable habitats of target species.** To obtain a map of suitable habitat for each species under the current conditions, the average value of the relative probability of occurrence calculated by the 10-fold cross-validation was translated into a Boolean habitat/non-habitat map. We applied the average of the 90% sensitivity threshold to minimize the false-negative fractions and to avoid underestimating the suitable habitat area[72,73].

Predicted suitable habitats may appear beyond the species' native ranges. For each species, we discarded projected suitable habitats if (1) they were beyond the current native regions recorded in the IUCN species database, or (2) if they are on a landmass that has not been connected to other landmass(es) with occurrence points since the last glacial maximum period. In this case, the threshold value for the paleo-coastline was set to −130 m below the current coastline[74], estimated by using seafloor topography data (ETOPO1)[75]. Although a few exceptional species with high dispersal ability may have the potential to go beyond their native range, to evaluate suitable habitat, we made rather conservative assumptions to minimize

commission errors (i.e., identification of suitable habitats in areas where a given species has never occurred owing to barriers or other biogeographic limitations).

**Estimation of dispersal ability of target species.** A species' ability to disperse and track the shifting climate is a crucial trait that determines its future potential for range shifts[76–78]. However, most of the previous global biodiversity models have not incorporated such realistic dispersal into their future projections and assumed only full-dispersal and no-dispersal scenarios, which has been criticized as unrealistic for most organisms[79]. The ability to simulate realistic dispersal has been limited by lack of knowledge about species' dispersal abilities or by technical constraints of the modelling. Recently, to incorporate more realistic dispersal, various approaches have begun to be used. Previous global scale studies[8,23,53] have incorporated a relatively simplistic approach based on the average dispersal rate of each target taxon. In this study, however, we incorporated variation of dispersal ability within taxonomic groups, which has not been considered in other global studies. The dispersal assumptions for each taxon in previous studies[8,23,53] were within the range of our estimated value, and no inconsistency occurred (Supplementary Fig. 9). Dispersal ability is strongly related to life-history traits, such as the dispersal syndrome and growth form of vascular plants[79] or the body mass and feeding habits of birds and mammals[80]. In this study, we collected information on life-history traits for all target species and used allometric equations based on these traits to estimate dispersal distance and generation length). Species-specific dispersal distance ($D$) between the present ($t_0$) and future ($t_1$) was calculated as (Eq.1)

$$D = d_g \times \frac{t_1 - t_0}{g}, \qquad (1)$$

where $d_g$ denotes dispersal distance per generation and $g$ denotes generation length.

For vascular plants, dispersal distance per generation was estimated from the formula based on Tamme et al.[79]. We adopted the group 5 formula in the "dispeRsal ()" function, which requires species-specific data for the dispersal syndrome and growth form. For the dispersal syndrome, we compiled the database from various sources (see source for Supplementary Fig. 9 in the Source Data file). For the growth form, we obtained the data from the IUCN Red List of Threatened Species[55]. Generation length was estimated based on Marbà et al.[81], according to growth form.

For amphibians, dispersal distance per generation was simply estimated from the allometric relationship between dispersal distance and adult body mass. We collected the dispersal distance data from Trochet et al.[82] and Smith and Green[83] and the adult body mass data from Trochet et al.[82] and Tacutu et al.[84]. Additionally, we collected snout-to-vent length and total length data from Trochet et al.[82] and AmphibiaWeb[85]. First, we modeled the relationship between snout-to-vent length (or total length) and adult body mass, and then we used this model to estimate adult body mass of species without this information. We then modeled the relationship between adult body mass and dispersal distance and used this model to estimate dispersal distance per generation for each species. Generation length was calculated based on life-history data (female maturity period, gestation period, and maximum longevity) derived from Tacutu et al.[84], according to the formula reported in the IUCN Red List of Threatened Species[55].

For reptiles, dispersal distance per generation was simply estimated from the allometric relationship between dispersal distance and adult body mass. We collected the dispersal distance data from various sources (Supplementary Table 3). Adult body mass data were derived from Myhrvold et al.[86] or estimated from the length–weight allometric equation reported by Meiri[87], based on snout-to-vent length data from Myhrvold et al.[86]. We modeled the relationship between adult body mass and dispersal distance, and then we used this model to estimate dispersal distance per generation for each species. Generation length was calculated based on life-history data (female maturity period, gestation period, and maximum longevity) derived from Myhrvold et al.[86] and Tacutu et al.[84], according to the formula reported in the IUCN Red List of Threatened Species[55].

For birds, dispersal distance was estimated from the formula reported by Hilbers et al.[80], which requires species-specific data for adult body mass and food habit (carnivorous or not). Adult body mass data were derived from BirdLife International[88], food habit data were derived from Wilman et al.[89], and generation length data were derived from BirdLife International[88].

For mammals, dispersal distance was estimated from the formula reported by Hilbers et al.[80], which requires species-specific data for adult body mass and food habit (carnivorous or not). Adult body mass data were derived from Pacifici et al.[90], food habit data were derived from Wilman et al.[89], and generation length data were derived from Pacifici et al.[90].

This method is currently the best available to incorporate variation in dispersal ability within taxonomic groups in a global scale study, although improvement is needed to represent all of the complex mechanisms related to dispersal[27].

**Future scenarios for land-use and climate changes.** We prepared two future scenarios of predictor variables: the mitigation scenario aimed to attain a radiative forcing in 2100 of around 2.6 W m$^{-2}$, whereas the baseline scenario did not consider GHG emission reductions. Each mitigation and baseline scenario has a corresponding land-use change scenario and climate change sub-scenario. We set the target years as the 2050s and 2070s.

To project future changes in suitable habitats, we used three hypothetical future scenarios (Supplementary Table 1). The first is the land-use change only (LU)

scenario, which assumed that land use will be changed according to either the mitigation or the baseline land-use scenario, while climate condition will remain constant as the current state. The second is the climate change only (CC) scenario, which assumed that land use will remain constant as the current state while climate will change according to the mitigation (RCP2.6) or the baseline (RCP8.5) climate change scenarios. The third is the land-use and climate change (LUCC) scenario, which assumed that both land use and climate will change.

Changes in land use under the mitigation and baseline scenarios were estimated by the AIM/CGE[25], a computable general equilibrium model representing the entire global economy. In the model, supply, demand, investment, and trade are described by individual behavioural functions that respond to changes in the prices of production factors and commodities, as well as changes in technology and preference parameters on the basis of assumed population, GDP, and consumer preferences. Land is represented as a part of the production functions, formulated as multi-nested constant elasticity substitution functions. The allocation of land by sector for 17 regions is formulated as a multinomial logit function to reflect differences in substitutability across land rent, and regional land use is further downscaled to high spatial resolution with the AIM/PLUM downscaling model[26] based on spatially explicit attainable yields. The spatial yields are aggregated and fed back into AIM/CGE. The AIM/CGE model has been widely used for climate change impact and adaptation studies[91,92].

In this study, we used future land-use variables generated by Fujimori et al.[29], which were based on five alternative socioeconomic conditions in the SSP framework[93]. The SSPs are based on five narratives describing how socioeconomic factors may change over the next century, considering changes in population, GDP, energy, emissions, and land use. These narratives are designed to span a range of futures in terms of the degree of difficulty for mitigation and adaption to climate change. Two of the SSPs describe futures in which challenges to adaptation and mitigation are both weak (SSP1: sustainability) or both strong (SSP3: regional rivalry). Two 'asymmetric cases' were designed, comprising a case in which strong challenges to mitigation are combined with weak challenges to adaptation (SSP5: fossil-fuelled development), and a case in which the opposite is true (SSP4: inequality). Finally a central case describes a world with intermediate challenges for both adaptation and mitigation (SSP2: middle-of-the-road). The SSPs employ a concept called scenario matrix architecture, which has a two-dimensional space comprising socioeconomic patterns and climate mitigation levels defined by radiative forcing levels. In this study, we used a radiative forcing level of 2.6 W m$^{-2}$ for the mitigation scenario under SSP1, 2, 4, and 5. We used a radiative forcing level of 3.4 W m$^{-2}$ for the mitigation scenario under SSP3, because there was no solution for 2.6 W m$^{-2}$ in SSP3. For the baseline scenario, we used the baseline condition in each SSP, which represents the absence of additional climate policy.

As land-use scenarios explicitly incorporate areas for bioenergy crops and afforestation for GHG mitigation activity, which did not exist in land-use data in the current condition, we had to merge these land-use types into the one that existed in 2005. Bioenergy crops were merged into cropland and afforestation into forests. These assumptions are rather optimistic and may not reflect the variation of environmental quality within each land-use type. For example, a land-use type's environmental condition may vary among management practices, which may cause large differences in local biodiversity (e.g., selection of tree species planted in afforestation[22], the time required for the forest to mature[94,95], and intensification of agricultural activity[34]). Although these differences may be less pronounced than the effect of land-cover change[34], we should note that our results may systematically underestimate the impacts of land-use change on biodiversity. Some aspects of agricultural intensification (fertilizer use, pesticide use, or machinery inputs) are endogenously determined in the AIM/CGE model, and all five SSP scenarios more or less estimate increases in agricultural productivity by technological development and irrigation expansion[33]. These effects of crop yield increments are reflected in the expansion rate of agricultural land in our model. However, if these increments are achieved through industrial agricultural practices (such as significant input of inorganic fertilizer or pesticide), it may add further risk for biodiversity loss[34]. On the other hand, the yield–biodiversity relationship is complex and multidimensional, and not all the technologies used to achieve yield increases have a negative impact on biodiversity[36,37]. Both the dimension (industrial or agro-ecological) of land-use intensification and its effect on biodiversity should be investigated in future research to address the large-scale impacts on global biodiversity that arise from climate change and mitigation activity.

Basically, the proportion of land that has changed from the current land use to another land use was higher in the baseline scenario (Supplementary Fig. 2). The total area of cropland (including bioenergy crops) and forest (including afforestation) was larger in the mitigation scenario than in the baseline scenario, whereas those of pasture and other natural land were smaller. Other main indicators for these scenarios are shown in Supplementary Figs 2–8.

Changes in climate under the mitigation and baseline scenarios corresponded to RCP2.6[30] and RCP8.5[31], respectively, which were used in the IPCC Fifth Assessment Report[9]. These representative concentration pathways are the GHG concentration pathways stabilizing radiative forcing at the end of the 21st century at ~2.6 and 8.5 W m$^{-2}$, respectively. We used future climatic variables based on five of the GCMs included in the Fifth Coupled Model Inter-Comparison Project (CMIP5) experiment: GFDL-CM3, HadGEM2-ES, IPSL-CM5A-LR, MIROC-ESM-CHEM, and NorESM1-M, which were downloaded from WorldClim[63]. The

mitigation scenario (RCP2.6) represents the most benign situation, with temperature likely to increase between 0.3 and 1.7 °C for 2081–2100, whereas the baseline scenario (RCP8.5) is the most extreme scenario, with increases between 2.6 and 4.8 °C projected for 2081–2100. All 19 bioclimatic variables for the future were calculated by using the same method as for the current climate (see section "Estimation of the current suitable habitats").

In this way, we prepared two cases of GHG emission pathways (baseline and mitigation) and two time periods (2050s and 2070s). Each of the four combinations of cases contains 35 scenarios: five LU, five CC, and 25 LUCC scenarios (Supplementary Table 1).

**Projection of suitable habitats under future scenarios.** The projected relative probability of occurrence for each target species was calculated by using the developed MaxEnt model and the prepared future scenarios explained above. Then the presence/absence map of each species was drafted by using the same methods used for the current conditions.

To identify potential future habitats constrained by dispersal distance (PFH-CDD) for the 2050s, the estimated dispersal distance for each species (see section "Estimation of dispersal ability of target species") was buffered. Then the areas of the PFH-CDD were calculated for each species. Next, for the 2070s, based on the PFH-CDD for the 2050s, the same procedure was applied again. This process was repeated for LU, CC, and LUCC.

In reality, land-use change and climate change occur simultaneously. Under such circumstances, losses or gains of suitable habitats occur not only due to land-use change or climate change, but also due to the combined effects of both. Therefore, it is necessary to evaluate the degree of contribution to those losses or gains by land-use change, climate change, or both. In this study, predicted lost suitable habitats are defined as pixels of suitable habitats under the current condition that become unsuitable under realistic future conditions (i.e., PFH-CDD in LUCC). Predicted gained suitable habitats are defined as pixels of non-suitable habitats under the current condition that become suitable under future LUCC conditions. Each of the predicted lost or gained suitable habitats were further divided into three elements: altered by land-use change, altered by climate change, and altered by the combined effects of land-use and climate changes. Suitable habitat lost by land-use change was defined as pixels in lost suitable habitats defined above and not in PFH-CDD in LU, but in PFH-CDD in CC. Suitable habitat lost by climate change was defined as pixels in lost suitable habitats defined above and not in PFH-CDD in CC, but in PFH-CDD in LU. Other pixels in lost suitable habitats are defined as suitable habitat lost by the combined effect of land-use and climate changes. Suitable habitat gain by land-use change was defined as pixels in gained suitable habitats defined above and in PFH-CDD in LU, but not suitable in PFH-CDD in CC. Suitable habitat gained by climate change was defined as pixels in gained suitable habitats defined above and in PFH-CDD in CC, but not in PFH-CDD in LU. Other pixels in gained suitable habitat are defined as suitable habitat gained by the combined effect of land-use and climate changes. The areas of each lost or gained suitable habitat were summarized for each of the five taxonomic groups and compared between mitigation and baseline scenarios in the 2050s and 2070s (Supplementary Fig. 1).

To test the significance of the difference in area of total lost or gained suitable habitat among numerous scenarios, we used generalized linear mixed-effects models (GLMM) with a gamma distribution of errors and log link function by using the "glmmTMB()" function of R package glmmTMB 0.2.3[96]. We fitted the model as a function of mitigation scenario (mitigation or baseline), year (2050s or 2070s), interaction term of scenario × year, SSPs, GCMs, and taxonomic groups, while using log(area of current suitable habitat) as an offset term. We fitted a random intercept of species identity to control variation among species in response to each scenario. In the model-fitting process, we added a very small value ($10^{-6}$) to the area of total loss, gain, and current suitable habitat.

## Data availability

The source data underlying Figs 1–4 and Supplementary Figs 2–9 are provided as a Source Data file. The additional data that support the findings of this study are available from the corresponding author upon request.

## Code availability

The code that support the findings of this study are available from the corresponding author upon request.

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

## Acknowledgements

We thank A. Mori and H. Matsuda of Yokohama National University; T. Shitara of Tsukuba University; K. Takano of the Nagano Environmental Conservation Research Institute; K. Fukasawa of the National Institute for Environmental Studies, Japan; S. Matsuhashi of the National Agriculture and Food Research Organization; and T. Kamo of the Forestry and Forest Products Research Institute, Forest Research and Management Organization for scientific and technical support and helpful comments. This research was supported by the Environmental Research and Technology Development Fund (S-14 and 2-1702) of the Environmental Restoration and Conservation Agency of Japan. In this research work we used the supercomputer of AFFRIT, MAFF, Japan.

## Author contributions

T.M. and Y.H. conceived the project; H.O. compiled the biodiversity database with substantial contributions from A.H.; H.O. analysed the biodiversity data with input from A.H., I.T., K.N., Y.K., N.T. and T.M.; T.H., S.F. and K.T. analysed the economy–land relationship under climate change policy by using AIM/CGE and AIM/PLUM; H.O. wrote the first draft of the paper. All authors contributed to writing and revising the text, with substantial contributions from T.H., S.F., K.T. and T.M.

## Competing interests

The authors declare no competing interests.
