## [Article File - Peer Review File · Nature Communications]

Reviewers' comments:

Reviewer #1 (Remarks to the Author):

The authors forecast future range loss of ~9000 species across taxonomic groups under, comparing the relative roles of climate change and land use change under RCP2.6 and RCP8.5 They provide a balanced, thoughtful, and clear presentation of the patterns of range loss. Overall, their findings are intuitive and make a strong case for mitigation by quantifying the benefits for biodiversity.

A number of aspects of modeling such a large number of species is quite impressive: Estimation of dispersal ability for ~9000 species, fairly comprehensive species-specific models selection, and a reasonable way to account for sampling bias (which is often ignored). I would've suggested using block CV stratified in env or geographic space to reduce the chance of spurious fits, but I wouldn't suggest redoing the whole analysis to deal with this. The rather coarse spatial resolution (60 km) and thinning presences to 1 per cell means that autocorrelation shouldn't be a huge deal.

It was unclear how high resolution land use data was obtained, as the future SSP products I'm aware of are only at quarter degree resolution.

How did you determine that land use caused range loss? Did you know the species habitat preferences?

Maxent doesn't predict absolute probability of occurrence as is mentioned in a few spots, but rather only relative occurrence rate, although this doesn't affect any of the results.

Reviewer #2 (Remarks to the Author):

This paper is a clear and well-described account of some developments and their application in an integrated assessment model, asking a clearly interesting and pertinent question. Would ambitious climate change mitigation benefit biodiversity or not?

I recognize the capacity of this and other IAMs to integrate a significant number of separate processes in a coherent framework, and I do not doubt that great care has been taken in this work. But I am convinced that, to be of value to the broader multi-disciplinary readership of nature communications, significantly more information would need to be provided. Recent IPCC reports have provided good illustrations, based on IAMs and other models, showing that the energy mix and the associated land use scenarios vary quite widely for different assumptions of mitigation pathways. Clearly, these assumptions must have a huge influence on the case being made in this paper, but in the given form of presentation, it does not become clear whether the stylized scenarios chosen for the analysis cover a significant part of this variability.

This is but one example for several aspects of this work where we can only trust the authors that suitable assumptions have been made - the reader cannot really develop herself an idea of what these assumptions could mean for the final conclusion of the study.

Not surprising, the final conclusions (bring climate and biodiv closer together) are a little obvious and currently discussed intensively already. At the least, I would have hoped that such an IAM study could be set up in order to identify the major fields where uncertainties exist.

In conclusion, I do not advise that this study is published in a generalist journal such as nature communications in its current form. The study setup would need to make more direct reference to the range of conditions that are currently described by ambitious mitigation scenarios than is currently the case here. And I have focused on this matter because of my own first-hand experience with it: I am quite convinced that there are other aspects of complexity that would require a more formal handling in this scenario analysis.

Reviewer #1 (Remarks to the Author):

The authors forecast future range loss of ~9000 species across taxonomic groups under, comparing the relative roles of climate change and land use change under RCP2.6 and RCP8.5 They provide a balanced, thoughtful, and clear presentation of of the patterns of range loss. Overall, their findings are intuitive and make a strong case for mitigation by quantifying the benefits for biodiversity.

A number of aspects of modeling such a large number of species is quite impressive: Estimation of dispersal ability for ~9000 species, fairly comprehensive species-specific models selection, and a reasonable way to account for sampling bias (which is often ignored). I would've suggested using block CV stratified in env or geographic space to reduce the chance of spurious fits, but I wouldn't suggest redoing the whole analysis to deal with this. The rather coarse spatial resolution (60 km) and thinning presences to 1 per cell means that autocorrelation shouldn't be a huge deal.

It was unclear how high resolution land use data was obtained, as the future SSP products I'm aware of are only at quarter degree resolution.

We used high resolution (0.5 degrees) land use data which were derived from AIM/CGE (Fujimori *et al.* 2012) and AIM/PLUM (Hasegawa *et al.* 2017), which have mentioned in 'Method' section. As we are urged to add more variation in SSPs according to other reviewer's comment, we replaced land-use data to the data publicly available (Fujimori *et al.* 2018, <https://doi.org/10.1038/sdata.2018.210>) [P27L555-P28L586]. In accordance with change in land-use data, some species failed from our analysis due to change in definition of land area. However, this change did not affect our final findings.

How did you determine that land use caused range loss? Did you know the species habitat preferences?

We determined the range loss caused by land use from combination of potential future habitats constrained by dispersal distance (PFH-CDD) on land-use change only (LU) scenario, climate change only (CC) scenarios, and the land-use and climate change (LUCC) scenarios. In this study, predicted lost suitable habitats are defined as pixels of suitable habitats under the current condition that become unsuitable under realistic future conditions (i.e., PFH-CDD in LUCC). Suitable habitat lost by land-use change was defined as pixels in lost suitable habitats

defined above and not in PFH-CDD in LU, but in PFH-CDD in CC [P30L620-P31L647, also see SI Fig. S1].

Maxent doesn't predict absolute probability of occurrence as is mentioned in a few spots, but rather only relative occurrence rate, although this doesn't affect any of the results.

Yes, indeed! We have added word 'relative' to some places [P26L519, P30L616].

Reviewer #2 (Remarks to the Author):

This paper is a clear and well-described account of some developments and their application in an integrated assessment model, asking a clearly interesting and pertinent question. Would ambitious climate change mitigation benefit biodiversity or not?

I recognize the capacity of this and other IAMs to integrate a significant number of separate processes in a coherent framework, and I do not doubt that great care has been taken in this work. But I am convinced that, to be of value to the broader multi-disciplinary readership of nature communications, significantly more information would need to be provided. Recent IPCC reports have provided good illustrations, based on IAMs and other models, showing that the energy mix and the associated land use scenarios vary quite widely for different assumptions of mitigation pathways. Clearly, these assumptions must have a huge influence on the case being made in this paper, but in the given form of presentation, it does not become clear whether the stylized scenarios chosen for the analysis cover a significant part of this variability. This is but one example for several aspects of this work where we can only trust the authors that suitable assumptions have been made - the reader cannot really develop herself an idea of what these assumptions could mean for the final conclusion of the study.

We appreciate to this comment. We have added the energy mix and land-use information together with socioeconomic variations (See SI Fig. S2-S8). These variations allow us to explore the different energy system, food consumption pattern and land-use system to attain the climate target. For example, in SI Fig. S3, we can clearly see that SSP1 which is characterized sustainable development has less energy demand, consumption in bioenergy and land-use change while SSP5 relying on largest economic growth would have strong increase in energy and bioenergy which eventually cause large scale bioenergy crop cultivation on land-use system. Interestingly, despite of these large uncertainty in the future scenario assumptions, we could robustly confirm our conclusion across different socioeconomic assumptions, which is climate change mitigation has net benefit in biodiversity.

Not surprising, the final conclusions (bring climate and biodiv closer together) are a little obvious and currently discussed intensively already. At the least, I would have hoped that such an IAM study could be set up in order to identify the major fields where uncertainties exist.

Indeed, this is excellent point. This kind of discussion has been made intensively (e.g., REFERENCE1,2,3). However, there are no studies that deal with land use scenarios and climate conditions consistently to our knowledge. For example, REFERENCE 4, 5 uses scenarios which has climate variations but their land-use assumptions are not consistent with overall climate change mitigations. For instance, low temperature stabilization scenarios in general requires large bioenergy crop land which can change the land-use pattern and it can eventually negatively affect to biodiversity, which can offset the benefit of climate change mitigation. These discussions have been made qualitatively but this study is the first paper which quantify these land use and climate change effect with consistent scenarios.

In conclusion, I do not advise that this study is published in a generalist journal such as nature communications in its current form. The study setup would need to make more direct reference to the range of conditions that are currently described by ambitious mitigation scenarios than is currently the case here. And I have focused on this matter because of my own first-hand experience with it: I am quite convinced that there are other aspects of complexity that would require a more formal handling in this scenario analysis.

Thank you for your helpful comments. To consider effect of variation in mitigation pathways, we added five scenarios varying socioeconomic assumptions characterized by Shared Socioeconomic Pathways (SSPs), which represent wide range of plausible socioeconomic development trends over the century [P28L568-L586] and can generate different characteristics of mitigation pathways. Our result was robust against variations in SSPs, although there was some difference by region [P15L244, P16L263-L267]. Furtherly, we added socio-economic information underlying our land-use scenario (e.g. energy-mix) to make our result understandable for multidiscipline leaders. And above all, we believe that it would be better to encourage multi-disciplinary research like this research to be open to the public widely at *Nature Communications* and to encourage wider discussion, and modified some sentence in this manuscript [P16L285-P17L288]. We believe these change made our paper suitable for *Nature Communications*.

Reviewers' comments:

Reviewer #1 (Remarks to the Author):

The authors have addressed my comments.

Reviewer #3 (Remarks to the Author):

Dear authors,

your work presents an impressive analysis of calibrating a large number of species distribution models in order to analyze the impact of climate change and land use change on biodiversity, as well as the relative contribution of both factors on biodiversity changes. While I am impressed by the amount of work, I am also left with a number of questions, the first of which is rather critical, while the others might merely be a matter of clarification.

I should first add that, contrary to one of the earlier reviewers, I do think this paper is relevant as well as comprehensible for a wide readership, and I think it fits the broad scope of Nature Communications well.

My most critical question is about significance. You present in Figure 1 the results for a large number of model runs, but are the biodiversity outcomes for MIT significantly different from BL? This needs to be analyzed in order to justify the title, and the main conclusions. Similarly, I would personally like to see that question answered for other differences as well: are Birds more affected than Reptiles? Are some SSPs preferable over others? (SSPs hardly seem to affect the biodiversity habitat).

I also miss some descriptive information on the species included. I understand the selection criteria, but I would like to know to what extent these are, for example, representative for IUCN threatened species, and the respective classes? In other words, are these relevant / important species?

Then, line 525 indicates that projected suitable habitats are discarded if they are beyond the current native regions. Yet, I think that climate change can do exactly that: change suitable habitats of a species beyond their native range. I would like you to justify this assumption (beyond the species dispersal distance, which I understand). Especially, I imagine it could affect the result in fig. 1 by only showing losses, as by definition this decision limits the gains, primarily.

A second question arises with respect to the incorporation of land use in the analysis. Following Fig S1 it is included twice: once in the species distribution model itself, and subsequently in terms of lost / gained habitat. First, I wonder whether this double counting isn't overly strict, with respect to land use, and second, it is not clear how land use translates into habitat. Following the description of Hasegawa et al (2017), land use is downscaled to 0.5 degree pixels, with a share of each of the 5 main land cover types. I don't see how this translates in the boolean habitat / non-habitat (such as described in line 627 and further). I also wonder if habitat is defined similarly for all species, or whether you differentiate between, say, forest species and other habitats, such as indicated in the IUCN database?

Finally, recent evidence (such as gathered for example in the PREDICTS database) emphasizes the importance of agricultural land use intensity, as well as other more subtle changes in the landscape, which are not included in your land use map. I understand the limitations of IAM models, but I feel you need to reflect on these processes.

Reviewer #4 (Remarks to the Author):

This paper addresses a great question which is the joint effects of land use change and climate change upon biodiversity, taking into account land use change associated with mitigation scenarios. Whilst this research has been painstakingly conducted, and contains interesting findings which are generally sound, I do not feel it reaches the standard required for a Nature journal, that is, in terms of being state of the art, and in terms of being sufficiently novel. The issue really is that there are already other publications already exist describing interactions between climate change and land use change based on a larger number of land use scenarios taken from a range of integrated assessment models, and also that a much smaller number of species has been modelled in MaxENT and at a coarser scale, than has been done in two other recent studies. Also there needs to be greater unpicking of what exactly is assumed in these mitigation scenarios. For example is afforestation, afforestation with native species, or with non-native species? That makes a very large difference to whether biodiversity benefits or not from this and it is not clear to me what is assumed about this. If it is assumed that native species are used and they are in fact not use, then negative implications for biodiversity may be overlooked. Or it could perhaps be that the results should have the caveat that they are only valid if the afforestation restores natural forest? Further, whilst it was creditable that the authors attempted to estimate dispersal for each species, the allometric equation approach described is not the right approach for considering range shifts, in particular for birds. Existing publications the authors might like to consult are Newbold, T. (2018) Proc Roy Soc B 285 (<https://royalsocietypublishing.org/doi/10.1098/rspb.2018.0792>) and Warren et al (2018) Science 360,791-795. There are also a number of earlier publications looking at the issue and the extent to which mitigation to 1.5C warming has a large land use footprint, and the paper needs to acknowledge the range of literature on this issue, and how different assumptions about the mitigation methods can lead to different implications.

Response to Reviewer 3

Reviewer #3 (Remarks to the Author):

Dear authors, your work presents an impressive analysis of calibrating a large number of species distribution models in order to analyze the impact of climate change and land use change on biodiversity, as well as the relative contribution of both factors on biodiversity changes. While I am impressed by the amount of work, I am also left with a number of questions, the first of which is rather critical, while the others might merely be a matter of clarification.

I should first add that, contrary to one of the earlier reviewers, I do think this paper is relevant as well as comprehensible for a wide readership, and I think it fits the broad scope of Nature Communications well.

My most critical question is about significance. You present in Figure 1 the results for a large number of model runs, but are the biodiversity outcomes for MIT significantly different from BL? This needs to be analyzed in order to justify the title, and the main conclusions.

Thank you for your constructive comment. To clarify this point, we performed a GLMM analysis of lost/gained suitable habitat and verified significant differences between the two scenarios (Wald test, $p < 0.001$). This analysis provided more support for our results, and thus we feel that there is no need to change our title and conclusions.

Similarly, I would personally like to see that question answered for other differences as well: are Birds more affected than Reptiles? Are some SSPs preferable over others? (SSPs hardly seem to affect the biodiversity habitat).

An additional GLMM analysis (which we mentioned in an earlier part of our response) enabled us to answer these questions in the affirmative for both dimensions. SSP1 was preferable over others with the least loss of suitable habitat and largest gain of suitable habitat. Birds are more affected than reptiles, with regard to mean loss of suitable habitat, which may be compensated for by their larger gain of suitable habitat that results from birds' high dispersal ability.

We added a description of the GLMM analysis on P38 L731-P39 L739 as follows: "To test the significance of the difference in area of total lost or gained suitable habitat among numerous scenarios, we used generalized linear mixed-effects models with a gamma distribution of errors and log link function by using the *glmmTMB* function of R package *glmmTMB* 0.2.3⁸³. We fitted the model as a function of mitigation scenario (mitigation or baseline), year (2050s or 2070s), interaction term of scenario \times year, SSPs, and taxonomic groups, while using $\log(\text{area of current$

suitable habitat) as an offset term. We fitted a random intercept of species identity to control variation among species in response to each scenario. In the model-fitting process, we added a very small value (10^{-6}) to the area of total loss, gain, and current suitable habitat.”

A description of the results is added on P13 L190-196 as follows: “Additionally, we found a significant difference in loss and gain of suitable habitat among SSPs (Tables 1, 2). SSP1 showed the smallest loss and relatively large gain in suitable habitat. SSP4 showed a relatively small loss and small gain of suitable habitat, whereas SSP3 and SSP5 showed a relatively large loss and large gain of suitable habitat. The difference in losses of suitable habitat among taxonomic groups was also significant: plants showed the largest loss, followed by birds, mammals, amphibians, and reptiles (Tables 1, 2).” Summary tables are included as table 1 and table 2.

A discussion was added on P21 L289-298 as follows: “Note that our results are highly dependent on assumptions regarding socio-economic conditions such as lifestyle³² or technological development related to energy efficiency³³ and biomass utility³⁴, which have the potential to affect future land-use changes. These aspects are partially expressed as a difference among SSP scenarios. Because SSPs involve changing several factors simultaneously and it is impossible to identify the factor that determines land-use change, several aspects of the land-use storyline in SSP1 (e.g., strong land-use change regulation, increasing crop yields, low animal-calorie shares, and low waste)¹⁸ may be the candidate feature that contributed to minimize conversion of forest and other natural land and prevent the loss of suitable habitats.”

I also miss some descriptive information on the species included. I understand the selection criteria, but I would like to know to what extent these are, for example, representative for IUCN threatened species, and the respective classes? In other words, are these relevant / important species?

In order to maintain accuracy of the model, we did not analyze species with a small number of occurrence records. As a result, our result tends to represent more common species than rare ones. By modeling a large number of common species, we believe that the representativeness of the ecosystem that species has inhabited is guaranteed. In the previous manuscript, we included the IUCN Red List rank of each species in the supplementary materials. In the revised manuscript, we added Supplementary Table 2, regarding the representativeness of each Red List rank, and added the following text to P32 L563-567: “As a result of data requirements, our target species have a high proportion of common species with low extinction risk and a low proportion of endangered species (Supplemental Table 2). Additionally, areas with intensively surveyed regions (e.g., North America and Europe) had a high proportion of modelled species (Supplemental Table 2).”

Then, line 525 indicates that projected suitable habitats are discarded if they are beyond the current native regions. Yet, I think that climate change can do exactly that: change suitable habitats of a species beyond their native range. I would like you to justify this assumption (beyond the species dispersal distance, which I understand). Especially, I imagine it could affect the result in fig. 1 by only showing losses, as by definition this decision limits the gains, primarily.

Our regional classification is very coarse, and the area defined as “native range” was large enough to cover the area potentially reachable within the year evaluated for most of the analyzed species. Although a few exceptional species with high dispersal ability may have the potential to go beyond their native range, we made conservative assumptions to minimize commission errors (i.e., identification of suitable habitats in areas where a given species has never occurred owing to barriers or other biogeographic limitations). We added the following sentence to P32 L582-P33 L586: “Although a few exceptional species with high dispersal ability may have the potential to go beyond their native range, to evaluate suitable habitat, we made rather conservative assumptions to minimize commission errors (i.e., identification of suitable habitats in areas where a given species has never occurred owing to barriers or other biogeographic limitations).”

A second question arises with respect to the incorporation of land use in the analysis. Following Fig S1 it is included twice: once in the species distribution model itself, and subsequently in terms of lost / gained habitat. First, I wonder whether this double counting isn't overly strict, with respect to land use, and second, it is not clear how land use translates into habitat. Following the description of Hasegawa et al (2017), land use is downscaled to 0.5 degree pixels, with a share of each of the 5 main land cover types. I don't see how this translates in the boolean habitat / non-habitat (such as described in line 627 and further). I also wonder if habitat is defined similarly for all species, or whether you differentiate between, say, forest species and other habitats, such as indicated in the IUCN database?

First, we must note that the effects of land use are not double counted in our analysis. In the first step, the relationship between species' occurrence and environmental variables (land use and climate) are analyzed by MaxEnt. Next, this relationship was used to translate the land use (and climate) map into a Boolean lean habitat/non-habitat map. These processes were conducted for each of the species, which reflect species-specific environmental requirements. To clarify the explanation, we changed “translation process from average value of relative probability of occurrence (continuous)” to “translated into a Boolean habitat/non-habitat map” in P32 L573.

Finally, recent evidence (such as gathered for example in the PREDICTS database) emphasizes the importance of agricultural land use intensity, as well as other more subtle changes in the landscape, which are not included in your land use map. I understand the limitations of IAM models, but I feel you need to reflect on these processes.

This is a really important point. Environmental conditions within each land-use type may vary due to different management practices, which may cause large differences in local biodiversity (e.g., further intensification of agricultural activity by using fertilizer or irrigation, selection of trees planted in afforestation, the time required for the forest to mature). However, this is currently the best available land-use data to address the one-way causal relationship of economy, land use, and biodiversity under various socio-economic pathways. Based on the reviewer's suggestion, we added the following sentence to P36 L661-L670: "These assumptions are rather optimistic and may not reflect the variation of environmental quality within each land-use type. For example, a land-use type's environmental condition may vary among management practices, which may cause large differences in local biodiversity (e.g., selection of tree species planted in afforestation²², the time required for the forest to mature^{80,81}, and further intensification of agricultural activity by using fertilizer or irrigation⁸²). However, this approach is currently the best available for addressing the one-way causal relationship of economy, land use, and biodiversity under various socio-economic pathways, and it still allows us to address the large-scale impacts on global biodiversity that arise from climate change and mitigation activity."

Response to Reviewer 4

Reviewer #4 (Remarks to the Author):

This paper addresses a great question which is the joint effects of land use change and climate change upon biodiversity, taking into account land use change associated with mitigation scenarios. Whilst this research has been painstakingly conducted, and contains interesting findings which are generally sound, I do not feel it reaches the standard required for a Nature journal, that is, in terms of being state of the art, and in terms of being sufficiently novel. The issue really is that there are already other publications already exist describing interactions between climate change and land use change based on a larger number of land use scenarios taken from a range of integrated assessment models, and also that a much smaller number of species has been modelled in MaxENT and at a coarser scale, than has been done in two other recent studies.

We believe that our study does have novel points compared to the two studies noted by reviewer #4. For example, Warren *et al.* (2018 *Science* 360,791-795) did not consider the effect of land-use change related to mitigation activity, as we did in our study. To the best of our knowledge, the interaction between land-use change and climate change was described by two previous studies (Newbold 2018 *Proc Roy Soc B* 285, DOI: <http://doi.org/10.1098/rspb.2018.0792> and Hof *et al.* 2018 *PNAS* 115(52), 13294-13299). However, their land-use predictions were generated by a single RCP-SSP combination and were not able to evaluate the differences among multiple socio-economic conditions that affect the magnitude of land-use changes. Therefore, we believe that our study is the first to evaluate the effects of the interaction of climate change and land-use change related to climate change mitigation activity under multiple socio-economic conditions.

Ours is also the first study to show that differences in socio-economic conditions can efficiently affect the synergy between climate change mitigation and the prevention of biodiversity losses. Our analysis also revealed that these differences were significantly related to the magnitude of land-use change, which is potentially driven by lifestyle or technological development related to energy efficiency and biomass utility, as well as regulation of land-use change. We emphasized these novel points in the revised manuscript, with citations of a range of additional literature on this issue.

Also there needs to be greater unpicking of what exactly is assumed in these mitigation scenarios. For example is afforestation, afforestation with native species, or with non-native species? That makes a very large difference to whether biodiversity benefits or not from this

and it is not clear to me what is assumed about this. If it is assumed that native species are used and they are in fact not use, then negative implications for biodiversity may be overlooked. Or it could perhaps be that the results should have the caveat that they are only valid if the afforestation restores natural forest?

In this study, we assumed afforestation with native species. Assuming non-native afforestation may emphasize the negative effect of land-use change from mitigation activity. The total area of afforestation was relatively small at a global scale (1.1–6.7% of the land surface), and it did not affect our conclusions when we checked in the preliminary analysis. However, we think this is an important point. Therefore, we added the following sentence to P20 L272-276: “However, note that our analysis may have overlooked the effect of land-use change arising from differences in environmental quality within each land-use type. For example, the negative impact of land-use change associated with GHG mitigation on biodiversity may become large if the afforestation is conducted by using non-native species, without any effort to restore the environment of natural forest^{21,22,}”

Further, whilst it was creditable that the authors attempted to estimate dispersal for each species, the allometric equation approach described is not the right approach for considering range shifts, in particular for birds.

There is a range of methods available to incorporate dispersal abilities in species distribution model studies. Currently, there is no single approach used to estimate species-specific dispersal ability. We used an approach that has been widely adopted in the recent literature to evaluate species-specific dispersal ability (e.g., Bateman *et al.* 2013; Hilbers *et al.* 2016), based on state-of-the-art scientific knowledge. Consistently, the dispersal assumptions for each taxon in previous studies (Warren *et al.* 2013, 2018; Newbold 2018) were within the range of our estimated value (Supplemental Fig. S9). Although our approach may still be insufficient to represent all of the complex mechanisms related to dispersal, we believe that it is currently the best available approach for incorporating variation in dispersal ability within taxonomic groups at the global scale.

To explain the dispersal estimation better, we added the following text on P33 L590-601: “However, most of the previous global biodiversity models have not incorporated such realistic dispersal into their future projections and assumed only full-dispersal and no-dispersal scenarios, which has been criticized as unrealistic for most organisms⁷⁴. The ability to simulate realistic dispersal has been limited by lack of knowledge about species’ dispersal abilities or by technical constraints of the modelling. Recently, to incorporate more realistic dispersal, various approaches have begun to be used. Previous global scale studies^{5,23,48} have incorporated a

relatively simplistic approach based on the average dispersal rate of each target taxon. In this study, however, we incorporated variation of dispersal ability within taxonomic groups, which has not been considered in other global studies. The dispersal assumptions for each taxon in previous studies^{5,23,48} were within the range of our estimated value, and no inconsistency occurred (Supplemental Fig. S9).”

Additionally, we added the following sentence on P33 L606-P34 L609: “This method is currently the best available to incorporate variation in dispersal ability within taxonomic groups in a global scale study, although improvement is needed to represent all of the complex mechanisms related to dispersal⁷⁴”

Existing publications the authors might like to consult are Newbold, T. (2018) Proc Roy Soc B 285 (<https://royalsocietypublishing.org/doi/10.1098/rspb.2018.0792>) and Warren et al (2018) Science 360,791-795. There are also a number of earlier publications looking at the issue and the extent to which mitigation to 1.5C warming has a large land use footprint, and the paper needs to acknowledge the range of literature on this issue, and how different assumptions about the mitigation methods can lead to different implications.

Both of those papers were cited in the previous manuscript. As mentioned in an earlier response, we believe that our study has sufficiently novel points compared to the two studies noted by reviewer #4. In response to this valuable suggestion, we added the following sentences to P21 L289-298: “Note that our results are highly dependent on assumptions regarding socio-economic conditions such as lifestyle³² or technological development related to energy efficiency³³ and biomass utility³⁴, which have the potential to affect future land-use changes. These aspects are partially expressed as a difference among SSP scenarios. Because SSPs involve changing several factors simultaneously and it is impossible to identify the factor that determines land-use change, several aspects of the land-use storyline in SSP1 (e.g., strong land-use change regulation, increasing crop yields, low animal-calorie shares, and low waste)¹⁸ may be the candidate feature that contributed to minimize conversion of forest and other natural land and prevent the loss of suitable habitats.”

We also we added the following sentences to P21 L304-P22 L309: “Currently, widely used land-use predictions are generated for single RCP-SSP combinations, which does not allow isolation of pure differences among socio-economic conditions²³. Further work that explicitly considers various socio-economic conditions and GHG mitigation options will need to specify the critical socio-economic factor necessary to achieve both GHG mitigation and biodiversity targets.”

Reviewers' comments:

Reviewer #3 (Remarks to the Author):

Dear authors,

You have addressed most of my comments appropriately, but I remain unconvinced by your remark about land use intensification.

First, some models already simulate changes in land use intensity. Maybe not yours, and it is for sure not the standard, but it is certainly possible (and already operational in other modelling teams - see for example Pouzols et al. (2014) and Kehou et al (2017).

Second, more importantly, you refer to land use intensity as the variation in environmental conditions. It is true that such conditions can and will vary, but my main concern is that over the past few decades (agricultural) land is has intensified across almost all world regions, and that under the SSP scenarios this trend continues. This indicates a systematic change in land use intensity in one direction (more intensive) which is associated with a decline in biodiversity. In other words, I think you systematically underestimate the effect of land use change on biodiversity by only looking at land use change associated with land cover change, while changes in land use intensities are omitted.

At the least I feel you should acknowledge this as a systematic bias over time, likely leading to an underestimation of biodiversity decline from land use change, not merely as variation across space.

Kehou et al (2017) Biodiversity at risk under future cropland expansion and intensification. *nature Ecology and Evolution* 1: 1129-1135.

Pouzols et al (2014) Global protected area expansion is compromised by projected land-use and parochialism. *Nature* 516 (383-386).

Response to Reviewer 3

Reviewer #3:

Dear authors,

You have addressed most of my comments appropriately, but I remain unconvinced by your remark about land use intensification.

First, some models already simulate changes in land use intensity. Maybe not yours, and it is for sure not the standard, but it is certainly possible (and already operational in other modelling teams - see for example Pouzols et al. (2014) and Kehou et al (2017)).

Second, more importantly, you refer to land use intensity as the variation in environmental conditions. It is true that such conditions can and will vary, but my main concern is that over the past few decades (agricultural) land is has intensified across almost all world regions, and that under the SSP scenarios this trend continues. This indicates a systematic change in land use intensity in one direction (more intensive) which is associated with a decline in biodiversity. In other words, I think you systematically underestimate the effect of land use change on biodiversity by only looking at land use change associated with land cover change, while changes in land use intensities are omitted.

At the least I feel you should acknowledge this as a systematic bias over time, likely leading to an underestimation of biodiversity decline from land use change, not merely as variation across space.

Kehow et al (2017) Biodiversity at risk under future cropland expansion and intensification. *nature Ecology and Evolution* 1: 1129-1135.

Pouzols et al (2014) Global protected area expansion is compromised by projected land-use and parochialism. *Nature* 516 (383-386).

Thank you for your constructive comment. To address this point, we added the following text in the discussion section (P20 L278-P21 L296): “Note, however, that our results may underestimate the impact of land-use change for two reasons. First, our models focus on land-cover change and are based on the assumption that the effect of land-use intensity within each land-use type does not affect species’ habitat suitability. For example, all of the five SSP scenarios more or less estimate an increase in crop yield in the future resulting from technological development and irrigation expansion³³, as reflected in differences in the expansion rates of agricultural land use in this study (i.e., if higher yield is expected, then

extension of agricultural land use could be limited). However, increasing crop yield through agricultural practices (e.g., significant input of inorganic fertilizer or pesticide) may add further risks for biodiversity loss³⁴ and require additional efforts for conservation³⁵. Although not all technologies that aim to achieve crop yield increases cause negative impacts on biodiversity^{36,37}, the lack of this factor in our species distribution model may have led to underestimated biodiversity losses from land-use change. As another example, the negative impact of land-use change associated with GHG emission reduction on biodiversity within the forest may become large if the afforestation is conducted with non-native species and without efforts to restore the environment of the original natural forest^{21,22}. These changes in habitat suitability arising from differences in environmental quality may cause systematic bias in our analysis.” We also added the phrase “environmentally sustainable intensification techniques” in P21 L321-322 to make our viewpoints clear.

The following text was added in the Method section (P37 L699-713): “Although these differences may be less pronounced than the effect of land-cover change³⁴, we should note that our results may systematically underestimate the impacts of land-use change on biodiversity. Some aspects of agricultural intensification (fertilizer use, pesticide use, or machinery inputs) are endogenously determined in the AIM/CGE model, and all five SSP scenarios more or less estimate increases in agricultural productivity by technological development and irrigation expansion³³. These effects of crop yield increments are reflected in the expansion rate of agricultural land in our model. However, if these increments are achieved through industrial agricultural practices (such as significant input of inorganic fertilizer or pesticide), it may add further risk for biodiversity loss³⁴. On the other hand, the yield–biodiversity relationship is complex and multidimensional, and not all the technologies used to achieve yield increases have a negative impact on biodiversity^{36,37}. Both the dimension (industrial or agro-ecological) of land-use intensification and its effect on biodiversity should be investigated in future research to address the large-scale impacts on global biodiversity that arise from climate change and mitigation activity.”